# CHOICES for sickle cell reproductive health: A protocol of a randomized preconception intervention model for a single gene disorder

**Diana J. Wilkie**[1]*, **Guettchina Telisnor**[1], **Keesha Powell-Roach**[2], **Andrea P. Rangel**[1], **Amelia L. Greenlee**[1], **Miriam O. Ezenwa**[1], **Agatha M. Gallo**[1], **L. Vandy Black**[1], **Alexandre Gomes de Siqueira**[3], **Brenda W. Dyal**[1], **Sriram Kalyanaraman**[4], **Yingwei Yao**[1]

1 College of Nursing, University of Florida, Gainesville, Florida, United States of America, 2 College of Nursing, University of Tennessee Health Science Center, Memphis, Tennessee, United States of America, 3 College of Engineering, University of Florida, Gainesville, Florida, United States of America, 4 College of Journalism and Communications, University of Florida, Gainesville, Florida, United States of America

* diwilkie@ufl.edu

## Abstract

Our long-term goal is to foster genetically informed reproductive health knowledge and behaviors among young adults with sickle cell disease (SCD) or sickle cell trait (SCT) with a web-based, tailored, multimedia intervention called CHOICES. CHOICES is designed to help young adults with SCD or SCT preconception to gain knowledge of genetic inheritance, specify their reproductive health intentions (their parenting plan), and engage in reproductive health behaviors concordant with their parenting plan. In a previous study, we found high acceptability of both the e-Book (usual care control) and CHOICES interventions. We also found sustained (24 months), significant effects on knowledge but not on behavior, most likely because half of the recruited group was not *at risk* for their children inheriting SCD. Hence, we propose an adequately powered randomized controlled trial with the CHOICES intervention and an e-Book control to compare their effects on genetic inheritance knowledge and *at-risk* reproductive health behaviors (immediate posttest and at 6, 12, 18, and 24 months). We will conduct subgroup analyses to provide insight into the baseline knowledge and behavior as well as the intervention effects in different demographic or acceptability groups. Given the scalability and low cost of CHOICES, if proved to be effective, it can reach the affected population at low cost.

## Introduction

About 100,000 Americans have sickle cell disease (SCD) [1] and 3.5 million Americans have sickle cell trait (SCT) [2]. Annual U.S. healthcare costs for SCD are an estimated $2.4 billion [3], where each year 2,000 infants are born with SCD, [1] and over 60,000 infants are born with SCT [4]. Advances in newborn screening and treatments have reduced SCD morbidity and mortality [5], they, however, have not reduced uninformed inheritance of SCD, despite support in minority and healthcare communities for this goal [6]. Pregnancy outcomes are

relevant data from this study will be made available upon study completion.

**Funding:** This research was made possible by Grant Number 1R01HG011927 from the National Human Genome Research Institute (NHGRI) and Grant Numbers U54HL090513, 1R01HL114404 and K01HL153210 from the National Heart, Lung, and Blood Institute (NHBLI) all part of the Institutes of Health (NIH). Its contents are solely the responsibility of the authors and do not necessarily represent the official views of the NIH, NHGRI, or NHBLI. The final peer-reviewed manuscript is subject to the National Institutes of Health Public Access Policy. The funders had no role in study design, data collection and analysis, decision to publish, or preparation of the manuscript.

**Competing interests:** D.J.W. is chairman and founder of eNursing llc, a company with no interests in the work presented in this manuscript. All other authors declare no conflicts of interests.

affected by insufficient or incorrect knowledge of one's own or partner's SCD/SCT status or SCD genetic inheritance [7–9]. Our long-term goal is to foster genetically-informed reproductive health knowledge and behaviors by young adults with SCD or SCT, which are single gene conditions that contribute to health disparities among minorities.

Responsive to RFA-HG-20-048, this study's focus is to evaluate a scalable and low-cost process for follow-up care after communication of genetic results to individuals with SCD and SCT. In the U.S., the annual incidence of SCD nationally remains fairly constant [10], but worldwide SCD is projected to increase from 304,800 newborns born annually to 404,200 in 2050 [11]. SCD is the most common inherited lethal blood disorder, and is a costly, single-gene disease [12]. For the years 2000 through 2006, 15,277 American infants were diagnosed with SCD through newborn screening in 44 states. With current human diasporas and massive migration, genetic sickle cell pools from Africa, Middle East, Mediterranean, Central and South America, India, and a growing Hispanic population (1 per 1,000–1,400 births) contribute to the U.S. SCD population [12].

One of every 500 African Americans has SCD, and one of every 12 African Americans has SCT [1]. SCT is usually asymptomatic but, as an autosomal recessive condition, persons with SCT are at risk for having children with SCD if their reproductive partners have SCT or SCD. A rough estimation based on prevalence of SCT and SCD shows that if no action is taken, 0.94 in 500 births among African Americans will exhibit SCD, very close to the reported statistic of 1 in 500 births. Thus, these rates suggest that the *at-risk* population is not currently engaged in systematic efforts to reduce the risk of their children inheriting SCD, an important scientific premise for the proposed study. If both reproductive partners have SCD, for each pregnancy, there is 100% chance of the child having SCD. If one partner has SCD and one has SCT or another hemoglobinopathy, for each pregnancy, there is 50% chance of the child having SCD. If both partners have SCT, for each pregnancy, there is 25% chance of the child having SCD. If only one partner has SCD or SCT and the other partner has normal hemoglobin (Hb A), for each pregnancy, there is 0% chance the child will have SCD. Examples of two behaviors needed for a person with SCD or SCT to reduce the risk of having a child with SCD include [1] talking with one's potential reproductive partner about sickle cell status and [2] seeking partners with Hb A. These behaviors, however, are not well-known or easy to accomplish for them or for any of the other 3 million Americans with one of the 10,000 other single-gene diseases (e.g., Huntington disease, Fragile X syndrome, cystic fibrosis, or muscular dystrophy) [13]. Study findings will provide evidence on how to mitigate uninformed inheritance of SCD worldwide and serve as an intervention model in follow-up to genetic counseling for other single-gene disorders.

In a series of four studies [14–17] we systematically established that the control and experimental interventions were relevant to the SCD community and that the participants were sufficiently engaged with the online computer delivery system. The CHOICES intervention significantly increased knowledge for young adults with SCD or SCT and this difference was sustained 24 months later. Intentions and behaviors were not significantly different between groups, most likely because half of the recruited group was not *at risk* for their children inheriting SCD. We concluded, therefore, that the proposed RCT should focus on individuals *at-risk* for having a child with SCD and whose parenting plan indicate that they want to avoid having a child with SCD and plan to have a child within the study period.

## Specific aims

Building on findings from the preliminary studies, we now propose a fully-powered RCT with refined methods and intervention enhanced with nudges and tailored boosters to compare

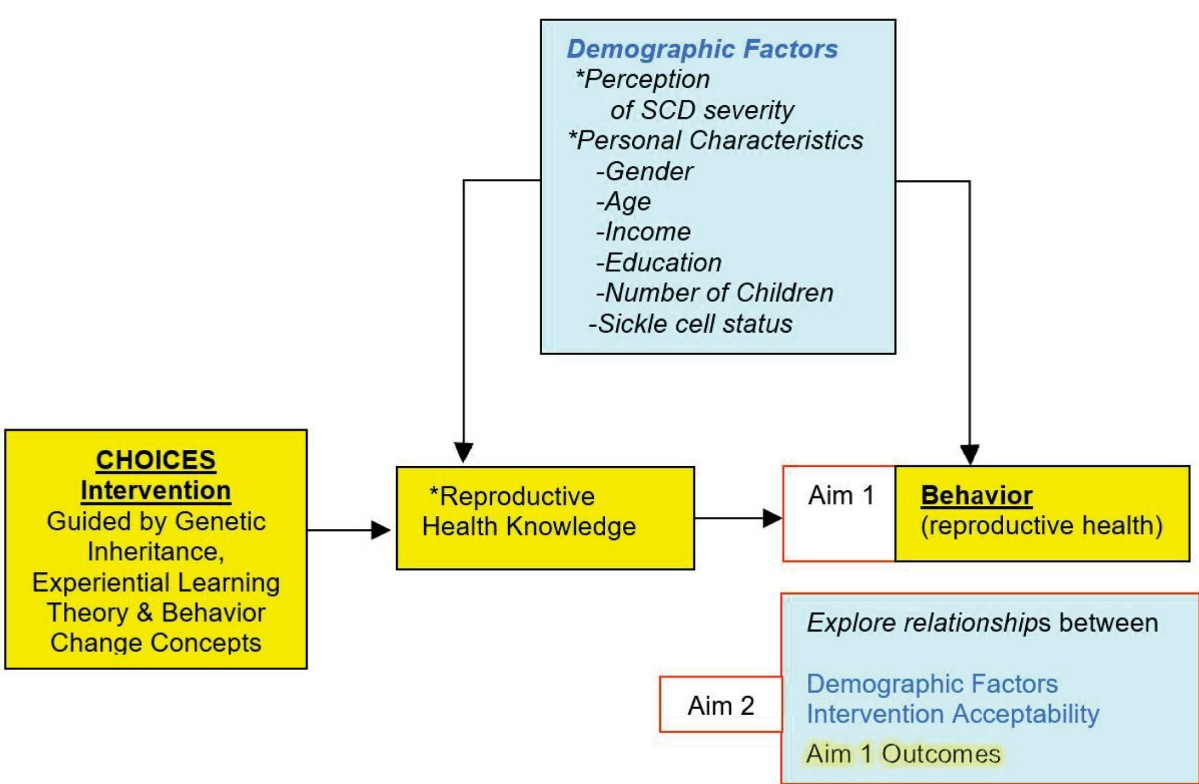

(Yellow boxes show the confirmatory study variables. Blue boxes show the exploratory study variables)

**Fig 1. Conceptual model and aims.**

effects of e-Book and CHOICES interventions on genetic inheritance knowledge (knowledge) and reproductive health behaviors (behaviors) (Fig 1). Using referrals from clinics, community-based organizations, transit advertising, and social media to recruit an adequately powered sample, we will use web-based data collection (SCKnowIQ—baseline, posttest, 6, 12, 18, 24 months) and intervention delivery (Fig 2). We will use multi-level regression analysis to examine the intervention effect on knowledge and behaviors (time *at-risk*). We will also explore predictors of outcomes. In a sample of 430 young adults (18–45 years of age, with SCD or SCT), specific aims are to:

**Aim 1.** Compare e-Book and CHOICES groups for effects on (a) knowledge (primary endpoint), and (b) *at-risk* behaviors (secondary endpoint) measured with the SCKnowIQ, over time (baseline, posttest, 6, 12, 18, 24 months). We hypothesize that (a) across all time points post baseline, there will be higher knowledge scores, and (b) lower percentage time with *at risk* behaviors (*at-risk* time) in the CHOICES than in the e-Book groups.

**Aim 2.** Explore the relationship between the demographic factors (e.g., gender, SC status), acceptability, and the Aim 1 endpoints (knowledge, behavior). We will conduct subgroup analyses to provide insight into the baseline knowledge and behavior as well as the intervention effects in different demographic and acceptability groups.

As a genetic counseling follow-up model, CHOICES may translate into informed parental decisions and preparedness for the consequences of their preconception decisions and informed inheritance of SCD.

**Sickle Cell CHOICES Study**

**Internet-based Study**

**Potential subject registers on the website**

Research Specialist (RS) **verifies screening information** according to eligibility criteria for all potential subjects registering on the website. RS contacts potential subject, arranges submission of **SCD or SCT status verification** and **signed consent, approves the subject to begin study**.

**Randomize**

**CONTROL Group**

Subject

- completes SCKnowIQ at Baseline
- Usual Care e-Book on computer
- completes SCKnowIQ (partial) at Posttest
- completes SCKnowIQ at 6, 12, 18, 24 months

**EXPERIMENTAL Group**

Subject

- completes SCKnowIQ at Baseline
- CHOICES on computer
- completes SCKnowIQ (partial) at Posttest
- CHOICES Nudges & Booster (6 & 12 months)
- completes SCKnowIQ at 6, 12, 18, 24 months

**Fig 2. Participant flow through the CHOICES sickle cell study.**

## Materials and methods

We are conducting a 2-year, randomized, longitudinal, repeated measures, controlled trial in 430 *at-risk* young adults with SCD (50%) or SCT (50%) to compare the effects of e-Book and CHOICES interventions on knowledge and behaviors across time (baseline, immediate post-test, 6, 12, 18, 24 months). We will provide boosters tailored to knowledge deficits at 6 and 12 months, but in the proposed RCT, we will also add monthly reinforcement nudges toward concordant behavior during the first 12 months. This clinical trial is registered at clinicaltrials. gov (NCT05292781), which exerted no control of this publication.

## Ethics statement

This study was approved by the Institutional Review Board (IRB) of the University of Florida to ensure ethical standards are upheld throughout the entire study.

## Setting and sample

The University of Florida (UF) will be the site for data collection, delivery of the intervention, and data analysis. Together, the UF Health Shands Hospital and UF Health Jacksonville Hospital serve approximately 933 patients with SCD. We will use Internet and remote technologies, which will allow participants to complete the study in their setting of choice (home, work, school, library, etc.) where Internet service is available. We will work with community based-organizations such as the Sickle Cell Disease Association of America (SCDAA) and its state-

based chapters, the African American sororities and fraternities, and county extension agents to identify young adults with SCD or SCT not only from UF but also from Florida and other clinics and communities across the nation. Based on attrition in our prior community-based studies of adults with SCD or SCT, we will recruit 506 participants. With a projected 15% attrition rate, we expect 430 participants to complete the study.

## Eligibility criteria

Participants are eligible to participate in the CHOICES study if they meet the following eligibility criteria: Confirmed diagnosis of SCD, SCT, or another hemoglobinopathy, able and intends to conceive a child, wants to avoid the risk of having a child with SCD, are able to speak and read English, are at risk for having a child with SCD, and are between the ages of 18–45 years.

## Recruitment

We will use online recruitment processes including registration on Clinical Trials.gov and the SCDAA's website. We will post recruitment information on Facebook, Instagram, TikTok, Twitter, Google+, other social media sites, and various Internet LISTSERVS to identify and recruit 506 young adults with SCD or SCT from across the U.S. (retaining 430). We used most of these processes and sites to recruit the Internet pilot study sample ($N = 115$ consented and $n = 107$ enrolled) that completed 24 months of follow-up at an 86% ($n = 92$) rate.

We will also distribute colorful posters and brochures in sickle cell clinics, at health fairs, university bulletin boards, churches, and other community-based sites. These posters differ from posters in our prior studies by intentionally utilizing language sensitive to individuals with SCD and SCT. These posters also utilize altruistic themes including descriptions on how their participation will make an impact. Additionally, our team members and Community Outreach Team will work directly with community-based organizations, healthcare networks, and churches to advertise the study within their networks, which are located throughout Florida and across the U.S. They and collaborators from clinics, SCD centers and organizations, and other community organizations (e.g., sororities, fraternities) will post recruitment flyers and encourage young adults to participate. These processes were effective in prior studies where we recruited 400+ participants mostly from IL but also from TX, NC, and CA. The participation of African American sororities and fraternities (collegiate and graduate chapters) is a new strategy for us and is likely to provide additional access to affected communities (college students, other young adults). African American sorority/fraternity members who are mothers and fathers of young adults will facilitate outreach to potential participants, especially those with SCT. These members live all over the U.S. and provide access not connected to healthcare settings. In our previous study, mothers who learned about the study were persistent in encouraging their young adult children to participate. Hence our ability to outreach to the sorority/fraternity members (who personally know their family members and friends affected by SCD or SCT) is likely to be successful.

We will also utilize transit interior bus advertisements in communities with large populations of individuals at risk for SCD or SCT. SCD advocates and influencers with substantial following on TikTok and Instagram will create videos to be released at different times promoting our study.

In summary, we will apply both previously tested and new strategies throughout the U.S., especially the Southern states that we intentionally did not sample in early studies so the powered efficacy study could access the states where nearly half of the individuals with SCD live [18]. Because Florida (FL) has a very large SCD population, we will leverage programs at the University of FL (UF) to identify and recruit participants with SCD or SCT. We will also

leverage a nationwide resource (available in every state) for community outreach, the Agricultural Experiment Stations, whose mission is to educate their communities. The FL Institute of Food & Agricultural Sciences (IFAS) leads this work in FL and their Extension Agents and State Specialists will connect us to other stations and post flyers and inform their communities about the study.

Finally, we will expand outreach to the newborn screening programs, which are now available in all 50 states, to identify efficient processes for potential participants to obtain verification of their SCT status. These innovative outreach approaches to communities for the purpose of identifying individuals with SCT will strengthen the generalizability of study findings.

## Screening and consent

As an Internet-based study, all screening and consent processes are completed via the website to which the individual is directed to all the recruitment materials. The individual completes the screening questions, reads the online consent form, calls a number to discuss questions with a qualified staff member, and signs the consent by typing his/her name and clicking a statement of consent. Before the individual is randomized, however, we require the research specialist (RS) to speak to the consenting person via a phone call and upload of an official healthcare record documenting SCD or SCT status. The RS who will assist individuals in these screening and consent processes will be trained in Good Clinical Practices for research and IRB and HIPAA procedures. The training includes role-play and retraining if the staff makes mistakes in the process(es).

Safeguards for pregnant women who are not a targeted population (participants could be pregnant on enrollment or during the study), include a behavioral intervention with important information related to reproductive health decisions. Therefore, risks to pregnant women are minimal, and no greater than what they might face from daily life.

## Randomization

Randomization to e-Book and CHOICES groups will be stratified by SCD and SCT to assure that each group is balanced on this variable. Within each stratum, permuted block randomization with a block size of four will be performed to ensure group balance. The statistician will conduct the randomization and the programmer will implement it via computer code. All other team members will be blind to the group assignment with no access to the electronic group assignment table within the software application.

## Retention and adherence

Important retention and adherence strategies include contacting participants for data collection and to update contact information every 6 months. We will send electronic birthday cards and birthday wishes via e-mail and text message. We will also obtain participants' addresses and telephone numbers and 3 other secondary contacts to allow us to track them during the 24-month study. We give incentives to motivate individuals to enroll in the study, maintain interest, and sustain participation for the 24-months of e-Book or CHOICES. All participants receive $20 for visits 1–4 as a small stipend for their time and to cover Internet access costs. We will also offer $30 for providing confirmation of their SCD or SCT status. At the 24-mo study visit, we will offer an incentive bonus of $40 for completing the 5th study visit; total stipend/incentive is $150 per participant, the same amount offered in our previous studies. Over 5 visits, participants are expected to take 230 minutes (3.8 hour) on average to complete the entire study over 24 months. We will make every effort possible to minimize participant

burden. In addition to the above incentives, we will use other general study procedures including: (1) texting, e-mailing, or calling participants to remind them that a study visit is due and to remind them when the visit window closes; and (2) updating contact addresses and phone numbers at each contact. In the pilot study, we contacted participants via text messages and social media platforms, a process that proved very effective for retention. We have ample evidence that these procedures enhance retention and result in low attrition and missing data rates, which leads to high study protocol adherence.

## Competition with other trials

As an innovative and unique reproductive health behavior trial, we expect that few other trials will pose potential competition. Our community-based recruitment plan, which was effective in our prior study, also minimizes this potential threat to our success.

## Available population

At UF, the UF Health Shands Hospital serves approximately 540 patients with SCD and the UF Health Jacksonville Hospital serves approximately 393 patients with SCD; 933 total patients with SCD. With an estimated 2019 population of 21.5 million that is 16% Black or African American, it is not surprising that Florida has a large population of patients with SCD, estimated at 8,374 to 14,236. Since we will recruit from across the U.S. for this Internet-based study, the available population of 100,000 individuals with SCD is more than adequate to meet our accrual goal of 506 recruited and 430 completed participants. Based on attrition in our prior community-based studies of adults with SCD or SCT, we will recruit 506 participants. With a projected 15% attrition rate, we expect 430 participants to complete the study.

## Sample power

Aim 1. In our 2-year longitudinal trial, the CHOICES group had an average knowledge score of 12.0 ± 3.0 and the e-Book group had an average knowledge score of 11.1 ± 3.1 at the end of the study. Based on this finding, the proposed sample of 430 can detect the intervention effect on participants' knowledge (primary outcome) with a power of 85%. In our exploratory analysis of at-risk participants wanting to be pregnant soon and avoid having a child with SCD, those in the CHOICES group were at risk 30% ± 37% of the time and those in the e-Book group were at risk 41% ± 44% of the time. Based on this preliminary finding, we estimated that the proposed study sample of 430 will provide 80% power to detect the intervention effect on the secondary outcome (time at-risk) with a two-sided Type I error rate of 0.05. Aim 2. The subgroup analysis proposed for Aim 2 is exploratory in nature. Our study is not powered to detect difference in knowledge and behavior outcomes between demographic groups or acceptability groups.

**Participant participation duration.** The duration of study participation is 24 months. The participant's flow of study activities appear in Fig 2.

## Procedures

The RS will develop Facebook pages for the study recruitment and retention processes. We will provide the potential participants with a link to the study website where they complete screening questions. RS receives confirmation of eligibility and sets up a time to discuss consent. Participants are sent a link to sign an online consent form, and have questions answered by the RS, who directs them to provide verification of SCD or SCT status. Once the RS receives the verification and approves the participants for participation, RS registers participants and

**Table 1. CHOICES sickle cell study data collection and intervention summary.**

| Study Event | Time Points (mo) | | | | | |
|---|---|---|---|---|---|---|
| | Baseline | Posttest | 6 | 12 | 18 | 24 |
| **SCKnowIQ** Computerized Questionnaires—Sickle Cell: | | | | | | |
| **Knowledge** Questionnaire | X | X | X | X | X | X |
| **Behavior** Tool | X | X | X | X | X | X |
| Demographic Questionnaire | X | | | | | |
| Intervention Acceptability | | X | | | | X |
| **CHOICES—Intervention** X initial; O nudges and booster tailored to most recent responses | X | | O | O | | |
| **e-Book—Intervention** X initial | X | | | | | |
| **Computer Acceptability Scale** | | X | | | | X |

participants are prompted to create a login and password on the CHOICES website. The software application randomizes the participants based on registration information after verified by the RS. The participants complete baseline measures and receive the CHOICES or e-Book based on group assignment. Table 1 lists a summary of the study data collection and intervention points, including boosters at 6 and 12 months and nudges monthly during the first 12 months of the study. For all intervention sessions, the experimental group reads or listens to the CHOICES intervention and the control group reads or listens to the usual care materials (e-Book) on the computer or mobile devices. At each follow-up visit, participants enter data with the SCKnowIQ data collection app (Table 1). Participants receive payments after each verified task via their selected method (mail, digital).

## Intervention

We created **CHOICES** (Fig 3), an innovative web-based software application to provide interactive information about SCD, its genetic inheritance, and reproductive health options for individuals with SCD or SCT. CHOICES is a multimedia intervention that provides targeted and tailored information for participants to overcome their misconceptions about their reproductive health options. Targeted to their gender and tailored to their knowledge, behavior during the previous 6 months, and parenting plan (i.e., intention to have child with SCD or not within the next 6 months), CHOICES presents information about options to help the participant achieve their parenting plan through reproductive partner selection based on sickle cell status, prenatal testing, ovum and sperm donation, advanced reproductive technologies, and options to avoid pregnancy or adopt, as well as information related to risks of pregnancy in the woman with SCD. The website will be housed on secure UF servers and accessible via computers or mobile devices (e.g., tablets, smart phones); device capture will inform plans for long-term dissemination.

We used Kolb's Experiential Learning Theory (ELT) [19] to organize the presentation of the CHOICES intervention material (Fig 3) and behavior concepts to frame the content. All the materials are written at an 8th grade level and based on evidence or best-practice facts or scripts for reproductive health in young adults with SCD or SCT. The facts or scripts in the booster intervention delivered at 6 months and 12 months are tailored to the participant's knowledge gap. We will modify the booster to focus more on the participant's risk behaviors that are not consistent with the current parenting plan and use nudges toward actualization of the concordant behavior that is needed to implement the parenting plan. The participant's responses on SCKnowIQ govern the specific health and behavior information provided in the booster and the participant's responses on the update queries associated with the nudges, which will be developed in the proposed study.

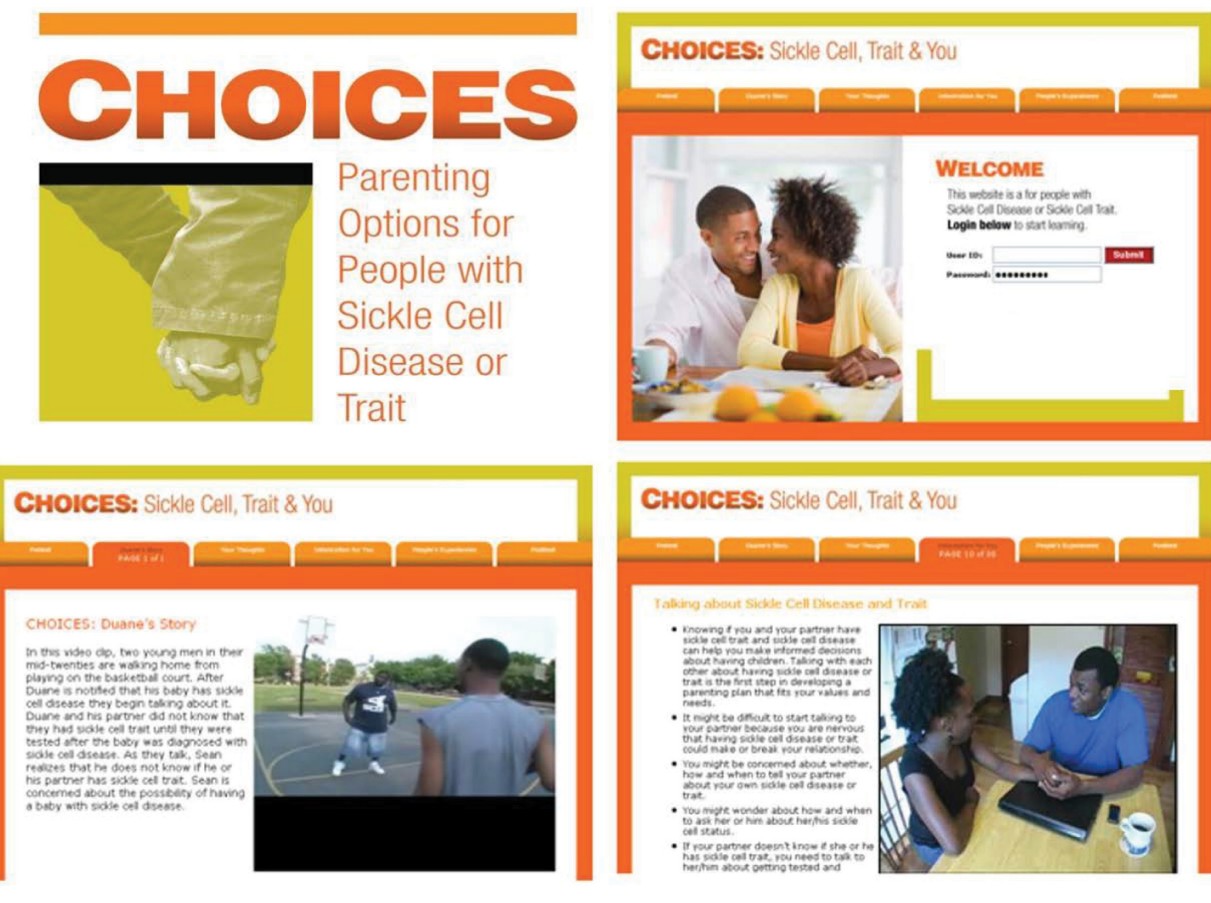

**Fig 3. Examples of CHOICES screens.**

Specifically, the CHOICES program reads the SCKnowIQ database for evidence of knowledge deficits about genetic inheritance of SCD and behaviors in the previous 1–6 months that are inconsistent with their parenting plan. CHOICES then presents information to target the knowledge gap and will be expanded to provide behavior guidelines, options, and role model nudges that help the participant implement the parenting plan, including any change in the parenting plan during the study. CHOICES respects the participant's parenting plan decision (there is no correct parenting plan other than the one the participant wants) and provides information about behaviors needed to enact the parenting plan. For this RCT, only participants stating they want to avoid having a child with SCD (their parenting plan at baseline) will be included, but based on the preliminary study at baseline, >90% reported this parenting plan and it was stable over 24-months.

The CHOICES program provides text, video clips, audio clips, graphics, cartoons, photos, and animations to help the participants overcome their knowledge gaps about SCD inheritance, available reproductive options, and state or role model the behavior steps to be taken to implement their parenting plans. There are extensive video clips of couples discussing their reproductive health choices to role model behaviors needed to implement their parenting plan, and how they feel about their choices and behaviors. For example, in one clip, a young couple breaks up after the young woman discloses that she has SCT and learns that her boyfriend also has SCT; she breaks up with him because she does not want any risk of having a baby with SCD and she would not consider other reproductive options, such as advanced

reproductive technologies or pregnancy termination, particularly since she wants a large family. Another couple discusses being adoptive parents because they do not want to risk the woman's health given that she has SCD. Video clips of these and other diverse scenarios bring reproductive health decisions to life serving to model the behaviors needed to implement the parenting plans and to supplement the text information. CHOICES includes skills training to promote mastery in performing the behavioral steps needed to implement the parenting plan, such as talking to one's partner, getting tested for SCT, determining risk given a couple's SCT or SCD status, selecting appropriate birth control methods, and adopting/fostering a child.

Therefore, the intervention is based on ELT to provide engaging and experience-based learning that also includes key elements important for concordant behavior. We will strengthen the behavioral component in the proposed study by adding reinforcement nudges and additional skills mastery components to the booster sessions. These additions will provide opportunities for skill development regarding concordant behavior that are needed to implement the parenting plan but were not performed in the previous 1–6 months. Boosters will also include positive reinforcement statements to recognize the participant's concordant behavior successes and encouragement for success and importance for concordant behavior. Participants completed the intervention in 71.8 ± 28.7 minutes; each booster in 8 minutes, which will be longer with the refinements that we plan to add based on preliminary findings.

The e-Book intervention contains information typically shared in clinical SCD care. It does not include other information related to the SCKnowIQ knowledge items or any of the behaviors needed to implement a parenting plan. The e-Book intervention will not include booster sessions, which differs from the preliminary study to minimize the repetition of genetic inheritance information, a key element of the CHOICES intervention. Although not as lengthy as CHOICES, the e-Book was completed at an average of 8.6 ± 2.8 minutes at baseline. It is sufficiently long, attractive, and engaging to retain participants randomized to the usual care control group; key factors for an adequate control condition. Interestingly, during the preliminary study and in interviews after the study, many e-Book participants commented that they were pleased to be assigned to the intervention group (even though they were not), which indicates that the control condition is sufficient for its intent—2-year longitudinal retention of the sample. Also, human contact is the same for both groups, and the time-on-task on the computer is unlikely to influence study knowledge and behavior endpoints.

## Acceptability of interventions

At the conclusion of the preliminary study's immediate posttest, participants in both groups were prompted by the computer to indicate what it was like to participate in the study and what they thought about the intervention program (Usability and Acceptability Instrument). Overall, they gave positive responses that included "great learning experience," "great program," "interesting," "helpful," and "informative." Although a few of the participants in the e-Book group saw the content as a "refresher," most participants in both groups found the content new and beneficial to their understanding and learning. Some of the participants went beyond focusing on their own learning and noted that the information would help stimulate awareness about SCD and SCT in individuals and communities. More of the CHOICES group than the e-Book group indicated that they "enjoyed" the learning and liked the various ways it was delivered including the presentation of the videos. Some participants clearly indicated that the computer was an easy, convenient, and "wonderful" way to convey this important information. As one participant said, "It beats classroom learning!" No negative comments were given about the use of computers. There were some suggestions about computer use, such as how to navigate the touch screen and desire to complete it at home on their own personal computers.

**Manipulation check for the fidelity of the usual care and CHOICES interventions.** Two assessments are made as intervention manipulation checks: 1) the delivery of the intervention (usual care e-Book or CHOICES) to participants and 2) the participant's take of the intervention. The manipulation check of the intervention delivery is assessed by calculating ratios for the number of intervention elements available to the participants and the actual number of intervention elements viewed by the participants. It is expected that this ratio will be 1 for most participants, indicating that they reviewed all the elements available to them. The 'take of the intervention' is measured by the gain in knowledge (study endpoint).

## Measures

We measure study process, outcome, descriptive and exploratory variables using a computer-based instrument that we named SCKnowIQ. SCKnowIQ includes items from previously validated instruments, items adapted from instruments that have been used in published population-based studies, and new items generated to measure concepts for which we were not able to find validated tools (Table 2). All instrument items were validated for the SCD and SCT populations and tested for reliability in the preliminary studies. The iterative methods used in the think aloud study with 20 participants produced valid study measures that are also reliable [16]. In total, participants take about 25 minutes to complete the measures at each time point. Items allow those not sexually active to give meaningful answers. Table 2 includes the validity and reliability information for each tool.

**Participant outcomes.** The SCKnowIQ tool includes 71 items, with many that are sex/gender specific [16]. The SCKnowIQ is a web-based app that was programmed based on many years of experience and a background in both current programming strategies and statistics. During the app's design and development, its usability was of high priority to avoid extraneous variability caused by users' differing experience with computers and their disparate physical and cognitive functions, which could contribute to increased measurement errors. A thorough usability test was conducted in the target population, and a few web pages were modified or improved by test outcomes [16]. We include managerial measures to ensure the data validity such as making sure that participants input data on time to avoid missing data or to ensure that data are collected on time.

**Knowledge.** The *Sickle Cell Reproductive Knowledge Questionnaire* portion of the SCKnowIQ measures knowledge of the genetic transmission of SCD and SCT, etiology of SCD, parenting options for people with SCD or SCT, types of contraceptives safe for people with SCD or SCT, and risks of complications during pregnancy for the woman with SCD. Responses are multiple choice options. Scores are totaled across all items, ranging from 0 to 17; higher scores reflect better knowledge than lower scores. Other details appear in Table 2.

**Table 2. SCKnowIQ computer data collection tool.**

| Concept Measured | Instrument Name | # Items | Internal Consistency | Test/Retest Reliability | Original Tool References |
|---|---|---|---|---|---|
| *Participant Outcome Variables* | | | | | |
| SC Knowledge Reproductive Knowledge | SC Reproductive Knowledge Questionnaire | 17 | .78 | .72 | Adapted from [20, 21, 24, 25] Validated in SCD/SCT [16] |
| Behaviors | Behavior Questionnaire | 8 | Not applicable (NA) | NA | Adapted from [20, 21, 26] Validated in SCD/SCT [16] |
| *Descriptive & Exploratory Variables* | | | | | |
| Demographic Characteristics | Demographic Questionnaire | 29 | NA | NA | Validated in SCD/SCT [16] |
| Acceptability | Usability and Acceptability Instrument | 20 | NA | NA | Validated in SCD/SCT [16] |

**Behavior:** *At-risk* **time.** The *Behavior Questionnaire* is an 30-item, sex/gender-specific measure of reproductive health behavior relevant to people with SCD or SCT [20, 21]. The items are part of the SCKnowIQ and focus on behaviors and outcomes during the past 6 months relevant to the success of their parenting plan: a) partner's sickle cell status; b) use of contraceptive methods; c) use of advanced reproductive technologies such as in vitro fertilization (IVF); d) pregnancy; e) use and outcome of prenatal testing; f) pregnancy outcome (e.g., pregnancy-induced changes, abortion, miscarriage, birth of an infant with SCD, birth of an infant unaffected by SCD or SCT, birth of an infant who is a carrier of SCT). The *at-risk* time of each participant during the 24-mo study will be obtained by aggregating the at-risk status outcomes of all visits. Based on their answers to the behavior questions (a-f) at each visit, we will assess whether they are at risk in the previous six months. A person is considered not at risk if their behaviors meet any of the following criteria: 1) their partner has Hgb A; 2) they always use effective birth control methods; 3) they use prenatal testing or advanced reproductive technology to ensure the baby is free of SCD (otherwise, a person will be considered *at risk*). The behavior items (f) will also allow us to calculate the number of children born during the study for descriptive purposes. The logical compatibility between behavior items and demographic items (SC status, number of children, etc.) in the same visit and across multiple visits of our previous studies and the participants' responses to cognitive interviews in the preliminary study [16] attest to their validity. Since all participants will be at risk at baseline, the posttest *at-risk* score ranges from 0% of the time at risk to 100% of the time at risk across the 4 posttest measures after the baseline visit.

**Descriptive and exploratory variables.** The *Demographic Questionnaire* includes items that are used to document the participant's gender, age, ethnicity, marital status, years of education completed, annual family income, SCD of self/relative, SCT of self and partner, use of hydroxyurea, severity of SCD, type of insurance, number of previous pregnancies, number of children, prior use of computers, and current access to computers. These data are collected as composite demographic data for the purpose of assessing the representativeness of the participants and analysis of the baseline characteristics of a heterogeneous sample of mostly African Americans. Other details about the tool appear in Table 2. The Usability and Acceptability Instrument is a 20-item tool designed to evaluate participants' perspective on the usability, acceptability and credibility of computer-based tools, such as CHOICES. The tool was validated in our preliminary studies as a method for identifying concerns about the study.

**Statistical analysis.** Data management and data analysis will be conducted by one of the MPIs and the study statistician, using statistical software R. All personnel with access to the data will be trained in the safety and security of privacy information during and after the study and will sign an oath of confidentiality prior to accessing any identifiable information. The data will be stored in a secure SQL database and will be exported to R for analysis. We will use an intention to treat approach, where all participants that are randomized will be included in our analysis.

Every effort will be taken in our software upgrade design to eliminate user errors; all data are entered directly by the participant using an interface tested with cognitive interview methods that informed the final interface design. Consistency checks will be built into the software so that inconsistent data (e.g., out of range or logically inconsistent with past data) will be flagged immediately and clarification from the users will be obtained. In addition, further and potentially more comprehensive consistency checks will be performed before data analysis. Inconsistent data points will be treated as missing and proper missing data processing (e.g., multiple imputation) will be applied. Based on our prior studies with this web-based software application, we expect the percentage of inconsistent data values to be very low (<0.5%).

For missing data, including those caused by participants missing visits, multiple imputation will be used to generate multiple completed datasets on which statistical inference will be performed and then aggregated. Missing at random assumption will be assessed and if necessary, sensitivity analysis will be performed using the pattern mixture model. Specifically, we will test various plausible selective missing mechanisms through post-processing of imputations to evaluate the robustness of our findings. We will consider a *p* value lower than 0.05 as statistically significant.

**Aim 1.**   Multi-level regression analysis will be used to analyze the longitudinal data to estimate the effects of the CHOICES intervention on knowledge and behavior, where random effect terms will be used to accommodate the within-participant correlation between repeated measures collected over time from each participant. We hypothesize that the CHOICES group participants will have significantly higher knowledge scores and lower *at-risk* behaviors compared with e-Book group participants. The knowledge outcome is the primary endpoint for determining success of the RCT, with the at-risk behavior outcome as secondary.

**Aim 2.**   We will compare the baseline knowledge and behavior by gender. We will also compare the intervention effects on different sexes by conducting regression analysis including gender, group assignment, and their interaction as predictors. Whereas this study is powered for detecting the intervention effects on the combined sample, not for determining difference between sexes, this valid subgroup analysis will provide important insight on potential difference in baseline characteristics and in intervention uptake by sex. We will also conduct exploratory analysis to investigate the relationship of SC status and other demographic characteristics such as age, education, among others and intervention acceptability with knowledge and behavior both at baseline and post-intervention. This exploratory analysis will provide insight on the baseline knowledge and behavior as well as intervention effects in different demographic groups and level of intervention acceptability, thus helping to identify groups most in need and most likely to benefit from the proposed intervention.

**Data sharing.**   Data and study-generated statistical code will be shared beyond the study investigators with an executed data use agreement. Study-generated intervention materials will be shared freely with the community via a website after the study results are published.

**Dissemination policy.**   Authors will include investigators participating in the study procedures, analysis, and manuscript writing. Professional writers will not be used.

**Data safety and monitoring.**   To ensure confidentiality, each participant is assigned a code number. All interviews and SCKnowIQ data are coded with the code number and linked to the participant's name in order to display tailored intervention with the participant's name. The participant's name is removed from the database when statistical analyses are conducted. The demographic information and other self-report outcomes will be entered at a secure web site on a secure application server with data written to a secure database server that is maintained by the UF Health Sciences information technology (UF IT) group. No data remain, even temporarily, on the computer used by the participant. Access to the database server is restricted to the lead statistician and the principal investigator (as backup) and the programmer. The two servers (application and database) are backed-up nightly, and there is fire- and water-secure, offsite storage of backups. The UF IT systems are highly controlled to minimize privacy or data breaches. All applications undergo a stringent security and privacy evaluation before being deployed on the IT managed severs.

## Discussion

Our study, the first of its type, has a strong scientific premise with many strengths to achieve the aims. With the exception of one recent study [22], there are virtually no other studies

except ours [14–17] published to date about young adults with SCD or SCT and their intentions to become parents. Our study is significant in addressing this need. We have a strong rigorous study design that builds on our prior studies and includes a longitudinal, randomized, repeated measures clinical trial with follow-up of knowledge and behavior for 2 years. This longitudinal design is needed to boost the intervention sufficiently to support concordant behavior and to allow participants sufficient time to engage in the behaviors needed to implement the parenting plan and then for us to measure behavior endpoints. A well-known theoretical framework, Kolb's ELT and nudges toward concordant behavior, guide our study. The research team is diverse, interdisciplinary, and exceptionally well qualified with expertise in SCD, cultural competence, strong engagement with community-based organizations that can facilitate recruitment, RCT design, methods and analyses, multimedia nudges, and targeted and tailored computerized interventions. Retention strategies are strong and will likely yield low attrition rates as in our past studies. The intervention is exceptionally innovative and provides both targeted (to group characteristics) and tailored (to individual responses) messages. Fidelity of intervention delivery is rigorous. Study instruments have acceptable psychometrics. Data analyses are well planned and take advantage of current approaches to repeated measures and missing data. If CHOICES produces successful concordant behavior in *at-risk* individuals with SCD or SCT, extension to studies with adolescent populations and international populations where millions of children are born annually will be warranted.

Although there are common challenges to building trustworthy relationships with participants from underrepresented minority groups, we have had success in doing so with a sample of young adults with SCD and SCT, more than 90% of whom are Black or African American. Our diverse team (5 Black, 3 Asian, 1 Latina, and 4 White) has experience in conducting research with the population and are continually mindful of respectful, genuine, and reciprocal approaches that foster and maintain trust. As in any study of health behavior, there is the possibility that self-reporting on the items may be under reported because of the personal nature of the items. We have the cultural expertise of several team members as well as a strong team of co-investigators, collaborators, advisors, and staff members, all of whom know the population and will vigilantly assist us to maintain our mindfulness of the cultural context of the participants.

We considered a health promotion intervention as an attention control condition rather than usual care. The e-Book is not "usual care" for individuals with SCT recruited from the community since they typically do not receive any genetic counseling under usual care. Hence, the e-Book provides new information for them. Even though our prior findings show that there were knowledge gains in the e-Book condition, which could diminish the group differences for the primary endpoint, we decided to retain the e-Book. We did so because our retention rate was high in the preliminary study, and it is not clear that a health promotion intervention not focused at all on SCD or SCT would exhibit comparable retention. Also, from an ethical perspective, we decided that the e-Book information on SCD/SCT should be available to all *at-risk* for their child inheriting SCD.

We considered randomization by State, instead of by individuals, as we did for the recently completed pilot study to avoid contamination that is possible if participants from the same state encounter each other and share information about their intervention. This type of contamination could lead to an underestimation of the intervention effect. However, our interviews of participants found that the interaction between participants from the same state was negligible. In addition, we observed substantial differences in characteristics of participants from different regions. Our experience, therefore, led to our decision to randomize at the individual level for the current RCT, because any potential loss of power due to the small risk and effect of contamination will be greatly outweighed by the large gain in study power of

individual randomization over cluster randomization with substantial differences between clusters. Self-report of reproductive health behaviors, which includes but is more than sexual activity, may not reflect actual sexual activity and is a limitation of the proposed study. We have taken steps to facilitate accurate capture of reproductive behavior, including use of online data capture, which is known for reduced reporting bias for sensitive information than an interview format [23]. We also used standard questions when possible, to reduce reporting bias.

Study findings will provide important evidence for primary prevention of the unbelievable suffering and the enormously high cost of healthcare associated with this inherited rare but lethal blood disorder. As a genetic counseling follow-up model, CHOICES may translate into informed parental decisions and preparedness for the consequences of their preconception decisions and reduced uninformed inheritance of single-gene diseases such as SCD. Considering the implications for SCD globally, the potential impact is enormous.

## Supporting information

**S1 Checklist.** *PLOS ONE* **clinical studies checklist.**
(PDF)

**S2 Checklist. SPIRIT 2013 checklist: Recommended items to address in a clinical trial protocol and related documents\*.**
(DOCX)

## Author Contributions

**Conceptualization:** Diana J. Wilkie.

**Data curation:** Yingwei Yao.

**Formal analysis:** Yingwei Yao.

**Investigation:** Diana J. Wilkie.

**Methodology:** Diana J. Wilkie, Yingwei Yao.

**Software:** Alexandre Gomes de Siqueira.

**Writing – original draft:** Diana J. Wilkie.

**Writing – review & editing:** Diana J. Wilkie, Guettchina Telisnor, Keesha Powell-Roach, Andrea P. Rangel, Amelia L. Greenlee, Miriam O. Ezenwa, Agatha M. Gallo, L. Vandy Black, Alexandre Gomes de Siqueira, Brenda W. Dyal, Sriram Kalyanaraman, Yingwei Yao.

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
