## [Decision Letter · Decision Letter 0]

13 Nov 2023

CHOICES for Sickle Cell Reproductive Health: A Protocol of a Randomized Preconception Intervention Model for a Single Gene Disorder

PONE-D-23-14304

Dear Dr. Diana J. Wilkie,

We’re pleased to inform you that your manuscript has been judged scientifically suitable for publication and will be formally accepted for publication once it meets all outstanding technical requirements.

Kind regards,

Ochuwa Adiketu Babah, FWACS, FMCOG

Academic Editor

PLOS ONE

Additional Editor Comments (optional):

Dear Authors,

Thank you for submitting your protocol for publication in PLOS One journal. The peer review process is complete, and we consider your protocol suitable for publication in its current form. However, We suggest you take into consideration suggestion made by reviewer 2 as this may be beneficial to you while preparing your final manuscript for publication following study completion.

All the best,

Dr Ochuwa Babah

Academic Editor,

PLOS one

“This research was made possible by Grant Number 1R01HG011927 from the National Human Genome Research Institute (NHGRI) and Grant Numbers U54HL090513, 1R01HL114404 and K01HL153210 from the National Heart, Lung, and Blood Institute (NHBLI) all part of the Institutes of Health (NIH). Its contents are solely the responsibility of the authors and do not necessarily represent the official views of the NIH, NHGRI, or NHLBI. The final peer-reviewed manuscript is subject to the National Institutes of Health Public Access Policy.”

Reviewers' comments:

Reviewer's Responses to Questions

**Comments to the Author**

1. Does the manuscript provide a valid rationale for the proposed study, with clearly identified and justified research questions?

Reviewer #1: Yes

Reviewer #2: Yes

2. Is the protocol technically sound and planned in a manner that will lead to a meaningful outcome and allow testing the stated hypotheses?

Reviewer #1: Yes

Reviewer #2: Yes

3. Is the methodology feasible and described in sufficient detail to allow the work to be replicable?

Reviewer #1: Yes

Reviewer #2: Yes

4. Have the authors described where all data underlying the findings will be made available when the study is complete?

Reviewer #1: Yes

Reviewer #2: Yes

5. Is the manuscript presented in an intelligible fashion and written in standard English?

Reviewer #1: Yes

Reviewer #2: Yes

6. Review Comments to the Author

You may also provide optional suggestions and comments to authors that they might find helpful in planning their study.

Reviewer #1: The potential siginifcance of CHOICES as a proposed research, if proven to be effective, is highly promising.

I am highly interested in the authors' study protocol and preliminary accepting of their project.

Reviewer #2: The protocol is well described and sufficiently detailed. The means of validating the acquired data in this web based protocol might need to be described further as internet bots could be engaged in certain tasks due to the repetitive nature of the tasks at different times.

Also, how to prevent the use of artificial intelligence in completing tasks during this protocol might need to be detailed.

7. PLOS authors have the option to publish the peer review history of their article (what does this mean?). If published, this will include your full peer review and any attached files.

Reviewer #1: **Yes: **Mena Abdalla

Reviewer #2: **Yes: **Dr. Hameedat Abdussalam

---

## [Editor Report · Acceptance letter]

30 Nov 2023

PONE-D-23-14304 

CHOICES for sickle cell reproductive health: A protocol of a randomized preconception intervention model for a single gene disorder 

Dear Dr. Wilkie:

I'm pleased to inform you that your manuscript has been deemed suitable for publication in PLOS ONE. Congratulations! Your manuscript is now with our production department. 

Kind regards, 

on behalf of

Dr. Ochuwa Adiketu Babah 

Academic Editor

PLOS ONE